# The Impact of GLP-1 RAs and DPP-4is on Hospitalisation and Mortality in the COVID-19 Era: A Two-Year Observational Study

**DOI:** 10.3390/biomedicines11082292

**Published:** 2023-08-18

**Authors:** Salvatore Greco, Vincenzo M. Monda, Giorgia Valpiani, Nicola Napoli, Carlo Crespini, Fabio Pieraccini, Anna Marra, Angelina Passaro

**Affiliations:** 1Department of Translational Medicine, University of Ferrara, Via Luigi Borsari, 46, I-44121 Ferrara, FE, Italy; grcsvt@unife.it; 2Department of Internal Medicine, Ospedale del Delta, Via Valle Oppio, 2, I-44023 Lagosanto, FE, Italy; 3Primary Care Department, Diabetes Unit of “SS. Annunziata” Hospital, Via Giovanni Vicini 2, I-44042 Cento, FE, Italy; v.monda@ausl.fe.it; 4Research and Innovation Section, University Hospital of Ferrara Arcispedale Sant’Anna, Via Aldo Moro 8, I-44124 Cona, FE, Italy; giorgia.valpiani@unife.it; 5Programming and Management Control Unit, University Hospital of Ferrara Arcispedale Sant’Anna, Via Aldo Moro 8, I-44124 Cona, FE, Italy; n.napoli@ospfe.it; 6Pharmaceutical Department, University Hospital of Ferrara Arcispedale Sant’Anna, Via Aldo Moro 8, I-44124 Cona, FE, Italy; c.crespini@ospfe.it (C.C.); a.marra@ospfe.it (A.M.); 7Pharmaceutical Care Department, Azienda Unità Sanitaria Locale della Romagna, Via Carlo Forlanini 34, I-47121 Forlì, FC, Italy; farmaciainterna.fo@auslromagna.it; 8Department of Internal Medicine, University Hospital of Ferrara Arcispedale Sant’Anna, Via Aldo Moro 8, I-44124 Cona, FE, Italy

**Keywords:** diabetes, GLP-1 RAs, DPP-4is, cardiovascular risk, inflammation

## Abstract

Novel antidiabetic drugs have the ability to produce anti-inflammatory effects regardless of their glucose-lowering action. For this reason, these molecules (including GLP-1 RAs and DPP-4is) were hypothesized to be effective against COVID-19, which is characterized by cytokines hyperactivity and multiorgan inflammation. The aim of our work is to explore the potential protective role of GLP-1 RAs and DPP-4is in COVID-19 (with the disease intended to be a model of an acute stressor) and non-COVID-19 patients over a two-year observation period. Retrospective and one-versus-one analyses were conducted to assess the impact of antidiabetic drugs on the need for hospitalization (in both COVID-19- and non-COVID-19-related cases), in-hospital mortality, and two-year mortality. Logistic regression analyses were conducted to identify the variables associated with these outcomes. Additionally, log-rank tests were used to plot survival curves for each group of subjects, based on their antidiabetic treatment. The performed analyses revealed that despite similar hospitalization rates, subjects undergoing home therapy with GLP-1 RAs exhibited significantly lower mortality rates, even over a two-year period. These individuals demonstrated improved survival estimates both within hospital and non-hospital settings, even during a longer observation period.

## 1. Introduction

Type 2 diabetes mellitus (T2DM) is known for being one of the most significant cardiovascular risk factors and has experienced a substantial increase in recent decades due to the rising burden of overweight and obesity [1]. Furthermore, in the past year, it has been strongly linked to the severe form of COVID-19 (coronavirus disease 19) [2], a genuine acute stressor for the diabetic population, resulting in generally high mortality rates [3,4]. Given that, according to various studies, the prevalence of diabetes in COVID-19 patients ranges from 5% to 36% [5], it is reasonable to assume that any additional cardiovascular risk factor can predispose individuals with both diabetes and COVID-19 to even worse clinical outcomes [6].

COVID-19 is characterized by a hyperinflammatory acute state [7,8,9], while also featuring low-grade inflammation that leads to chronic complications. The simultaneous presence of these two conditions likely explains why patients with T2DM suffer from higher mortality rates compared to other subjects. Another point connecting diabetes with coronaviruses is that both SARS-CoV (severe acute respiratory syndrome coronavirus) and SARS-CoV-2 (the causative agent of COVID-19) can enter respiratory tract cells by exploiting the angiotensin-converting enzyme 2 (ACE2) and binding to the spike protein on the virion surface [10]; the effects of this enzyme are multifold, and it also has a major role in micro and macrovascular complications in subjects with diabetes [11].

In contrast, MERS-CoV (Middle East respiratory syndrome), the second member of the coronaviruses family, uses a different receptor for cell penetration, known as dipeptidyl-peptidase-4 (DPP-4) [12]. This receptor can degrade glucagon-like peptide-1 (GLP-1), a hormone produced by L cells in the distal ileum in response to glucose passing through the intestines, a phenomenon called the “incretin effect” [13]. The expression of GLP-1 receptors is not limited to the gastrointestinal tract; they have been found in other systems such as the central nervous system, respiratory system, and cardiovascular system. These receptors also regulate glucose homeostasis in non-diabetic patients, promoting insulin secretion and inhibiting glucagon production [14]. When these mechanisms malfunction, as is the case in T2DM, uncontrolled glucose homeostasis leads to chronic inflammation [15].

Diabetes and inflammation are closely intertwined regardless of other diagnoses, and some antidiabetic drugs have been developed to provide enhanced cardiovascular protection through their anti-inflammatory effects, independent of their glucose-lowering actions [16]. This effect of the novel classes of diabetes medications is one potential mechanism to consider when examining their cardiovascular benefits, even though evidence on this matter is still limited, necessitating further clinical trials to explore this aspect of diabetes.

COVID-19 shares with diabetes the ability to intensely stimulate the immune system. However, the infection caused by SARS-CoV-2 can exploit this function more rapidly compared to diabetes, which requires more time. Based on this, COVID-19 can be considered, in all respects, an acute stressor for the body, like a major adverse cardiovascular event (MACE). Other anti-inflammatory drugs have previously been suggested to be protective against COVID-19. Orally delivered DPP-4 inhibitors (DPP-4is) (such as sitagliptin, vildagliptin, saxagliptin with mimetic inhibition mechanisms, alogliptin, and linagliptin with non-mimetic inhibition) and GLP-1 receptor agonists (GLP-1 RAs) with daily (exenatide, lixisenatide, liraglutide) or weekly (semaglutide, exenatide LAR, dulaglutide) subcutaneous administration [17,18] or once-daily oral administration (semaglutide) are among these potential treatments [19].

In this article, we delve into the role of antidiabetic drugs in relation to SARS-CoV-2 infection as an acute stressor, with a particular focus on individuals chronically treated with GLP-1 RAs or DPP4-is. Our work has three main objectives: (a) comparing, in a “real-world” setting, the hospitalization rates of various medical conditions in T2DM patients on home therapy with GLP-1 RAs or DPP-4is (either alone or in combination with other antidiabetic drugs) versus those on home therapy with different antidiabetic agents and/or insulin; (b) comparing the length of hospital stays between the two groups; and (c) calculating the mortality rates (all-cause and COVID-19-related mortality) of subjects with diabetes in the different groups. The observation period was extended to a second year following the initial observation.

## 2. Materials and Methods

### 2.1. Study Design

This is a retrospective, multi-center, non-interventional, observational cohort study. We enrolled a total of 76,764 patients from hospitals in the districts of Ferrara and Romagna (the University Hospital of Ferrara (Coordinating Centre), as well as the Ferrara and Romagna Local Health Units (LHUs)). Additional details about the participating centers can be found in Appendix A.

The databases used for analysis included the demographic database, pharmaceutical database (containing data related to dispensed drugs, categorized by the Anatomical–Therapeutic Chemical [ATC] codes FED for “Farmaci a erogazione diretta” and AFT for “Assistenza Farmaceutica Territoriale”), and hospitalization database. Hospital discharge cards (HDCs) were used to track internal transfers between operating units, providing information such as admission, transfer, and discharge dates and times, admission diagnoses, and previous history of major adverse cardiovascular events (MACEs), including non-fatal myocardial infarction (MI), non-fatal cerebrovascular accident (CVA), heart failure (HF), malignant dysrhythmias (MD), and cardiac shock (CS).

Clinical diagnoses were classified using the International Classification of Diseases, 9th Revision, Clinical Modification (ICD-9-CM), while mortality data were collected daily from the ReM (Relevation of Mortality) service for the Emilia Romagna Region. Information about individuals with diabetes in each region was gathered in collaboration with local diabetology units.

SARS-CoV-2 infection was confirmed through nasopharyngeal swabs with virus-specific RNA detection and amplification using real-time polymerase chain reaction assays (RT-PCR). Hospitalization was categorized as “COVID-19-associated” or “-related” based on diagnoses listed on the hospital discharge card for coronavirus infections, as communicated by the hospital assistance service of the Emilia Romagna region in March 2020.

The inclusion criteria consisted of two factors: (a) age, individuals aged 18 or older; and (b) diabetes-specific drugs used directly for diabetes treatment following the Anatomical Therapeutic Chemical (ATC) classification, A10. The observation period spanned two years, from January 2020 to December 2021.

The use of an anonymous unique numeric code ensured full compliance with the European General Data Protection Regulation (GDPR) (2016/679). The analysis was performed exclusively on anonymized data, thereby adhering to privacy regulations. Results were presented only in aggregated form, preventing attribution to any single institution, department, doctor, or individual prescribing behavior. The study was conducted in accordance with current legislation for retrospective studies. According to the Data Privacy Guarantor Authority (the General Authorization for personal data treatment for scientific research purposes—n.9/2014); informed consent was not required due to organizational constraints. The study adhered to Italian law on observational studies, with notification to and approval from the ethics committee of each participating entity.

### 2.2. Demographic Data

The University Hospital of Ferrara (UHF) and Ferrara Local Health Units (LHUs) collectively cover an area inhabited by 345,503 people (45,387 of whom are under 18 years old, and 300,116 of whom are older), while the region served by the Romagna LHUs has a population of 1,125,574 individuals (174,618 under 18 years old and 950,956 older), making a total of 1,471,077 residents.

Regarding adult T2DM patients, the Ferrara district reported 17,797 individuals (constituting 5.2% of the population), and the Romagna area had 59,327 cases (equivalent to 6.2%). This resulted in a combined diabetic population of 76,764 patients, accounting for 6.1% of the adult population. It is worth noting that these figures align closely with the national average, considering the prevalence of diabetes in Italy (which stood at 5.9% for both female and male subjects in 2020; source: www.istat.it, accessed on 1 January 2023). This demonstrates the significant consistency of our region’s data with the country’s overall statistics. Additional details concerning the demographic breakdown of subjects can be found in Appendix A.

We divided the overall population with T2DM into several subgroups based on their respective home antidiabetic therapies. Additionally, the cohort of patients who were hospitalized was further divided into two subgroups, categorized by the reason for their hospital admission (COVID-19-related or other reasons).

The term “non-COVID-19-associated-related hospitalization” refers to admissions prompted by various causes requiring hospital treatment, excluding SARS-CoV-2 infection. Examples of such causes include cardiovascular events, routine or emergency surgeries, infections, respiratory insufficiency, and more.

The hospitalized population with T2DM was further characterized and classified based on their home antidiabetic therapy and their history of major adverse cardiovascular events (MACEs). We sought to identify differences between groups in terms of individual MACEs (such as non-fatal myocardial infarction, MI; non-fatal cerebrovascular accident, CVA; heart failure, HF; malignant dysrhythmias, MD; cardiac shock, CS), 2-point MACEs (non-fatal MI and non-fatal CVA), 3-point MACEs (non-fatal MI, non-fatal CVA, and HF with or without CS), and 4-point MACEs (non-fatal MI, non-fatal CVA, HF with or without CS, and MD).

### 2.3. Statistical Analysis

Data analyses were carried out using IBM SPSS Statistics version 26.0 (IBM Corporation). The normality of the distribution of continuous variables was assessed using the Shapiro–Wilk test. In case of normal distribution of data, continuous variables were presented with their mean and standard deviation (SD), while in case of non-normal distribution, with their median value and interquartile range [1Q 3Q]. Categorical data were presented as total numbers and percentages (%).

Differences between groups were examined in terms of age, sex distribution, length of stay, mortality (both in-hospital and cumulative deaths within the two-year observation period), and antidiabetic treatment. Percentages were compared using the chi-square test, Fisher’s exact test, or Yates’ correction if necessary. Continuous data were assessed using Student’s *t*-test or the Mann–Whitney test as appropriate.

Box plots were employed to compare the lengths of stay among different groups of inpatients. Additionally, chi-square tests were conducted for risk estimates, and one-versus-one analyses among the primary antidiabetic treatments, with computation of relative odds ratios (ORs) and 95% confidence intervals (CIs). These ORs and 95% CIs were calculated using an unadjusted logistic regression model, with the need for hospitalization, in-hospital deaths, and cumulative deaths as dependent variables, and age, sex, 4-point MACE events, and antidiabetic treatments as independent variables. Survival curves were generated using the Kaplan–Meier method and compared between various subgroups using the log-rank test. A significance level of *p* < 0.05 was considered statistically significant.

Throughout the development of this article, STROBE (Strengthening the Reporting of Observational Studies in Epidemiology) guidelines were adhered to in each phase.

## 3. Results

### 3.1. Antidiabetic Prescriptions

We categorized the population with T2DM based on their respective home antidiabetic treatments. As anticipated, a significant proportion of the population (30,238 individuals, accounting for 39.4% of the total) were undergoing therapy exclusively with metformin. Furthermore, 14,739 individuals (19.2%) were treated with either insulin or insulin secretagogues. An additional breakdown of the population included 2037 individuals (2.4%) utilizing a combination of DPP-4 inhibitors (DPP-4is) and metformin; 1169 individuals (1.5%) on DPP-4is alone; 1095 individuals (1.4%) taking DPP-4is along with insulin or insulin secretagogues like glimepiride, glyburide, or glipizide; 910 individuals (1.2%) using GLP-1 receptor agonists (GLP-1 RAs) alongside metformin; 190 individuals (0.2%) on a regimen of GLP-1 RAs in combination with insulin or insulin secretagogues; and 127 individuals (0.2%) solely on GLP-1 RAs. Moreover, 26,259 individuals (34.2%) were on various other drug combinations or alternative medications such as SGLT-2 inhibitors, acarbose, or thiazolidinediones (refer to Table 1).

### 3.2. Population Characteristics

Within the primary cohort of individuals with T2DM, a total of 2910 required hospitalization, representing 3.8% of the cohort. Among these hospitalizations, 1922 cases (66.0%) were attributed to COVID-19, while the remaining 988 cases (34.0%) were due to other reasons. Table 1 provides a comprehensive overview of the distinctions between the two groups of subjects who were hospitalized for COVID-19 or alternative reasons. The *p*-values resulting from the comparison of COVID-19 and non-COVID-19 inpatients are presented.

Substantial differences were observed in terms of age; COVID-19 patients were relatively younger, with an average age of 73 ± 13 years compared to 74 ± 13 years for non-COVID-19 inpatients (*p* = 0.05). This age difference was particularly pronounced within the subgroup of individuals aged over 80, where those hospitalized for non-COVID-19 reasons were generally older (*p* = 0.05). Regarding sex distribution, males were more prevalent in both subgroups of inpatients.

In-hospital mortality rates demonstrated significant variation; COVID-19 patients experienced notably lower mortality, with rates of 21.9% compared to 32.2% for those hospitalized for other reasons (*p* < 0.001). Cumulative mortality data further supported this trend, with COVID-19-related mortality at a lower level of 28.9% in contrast to 42.8% for non-COVID-19-related reasons (*p* < 0.001).

The same categorization (subjects with T2DM hospitalized for COVID-19 versus those hospitalized for other reasons) was maintained in the second section of Table 1, wherein we assessed the differences in terms of individual MACEs and subsequently, 2-point, 3-point, and 4-point MACEs, as elaborated above. Noteworthy differences between the two groups emerged for non-fatal cerebrovascular accidents (CVA) (8.2% vs. 5.8%, *p* = 0.023), heart failure (HF) (12.7% vs. 9.6%, *p* = 0.022), and cardiac shock (CS) (0% vs. 0.5%, *p* = 0.019). COVID-19 inpatients presented generally lower rates of 4-point MACE (16.4% vs. 21.8%, *p* = 0.004). Aside from CS, where subjects admitted for other reasons did not experience this outcome, those admitted due to COVID-19 generally exhibited a lower occurrence of MACEs. This trend persisted across 2-point, 3-point, and 4-point MACE analyses.

Regarding antidiabetic treatments, no substantial differences were encountered in the comparisons between groups.

In Appendix A, we have presented the prevalence of each individual MACE, categorized based on patients’ antidiabetic home treatment. Given the notably low percentage of occurrences for each MACE within all subgroups, no statistical analysis or intergroup comparison was deemed relevant.

### 3.3. Antidiabetic Drugs and Outcomes

In Figure 1, we direct our attention to the primary treatment groups discussed earlier, excluding the subgroup treated with other drugs or combinations of drugs. For each treatment group, we calculated the respective percentages of hospitalizations (attributed to COVID-19 or other reasons), in-hospital deaths (separating COVID-19-related and other reasons), and cumulative deaths (encompassing both hospitalized and non-hospitalized subjects).

Regarding the need for hospitalization (Figure 1A), the breakdown is as follows: among subjects treated with metformin, 1.8% required hospitalization due to COVID-19, and 0.8% for other reasons; within the group receiving insulin/insulin secretagogues, 3.2% were hospitalized for COVID-19, and 1.9% for other reasons. For those treated with GLP-1 Ras, 2.8% were admitted due to COVID-19, and 1.5% for other reasons, while among the subjects treated with DPP-4 inhibitors (DPP-4is), the respective percentages were 3.6% for COVID-19-related hospitalization and 2.0% for other reasons.

The analysis pertaining to mortality yielded different results; among all treatment groups, patients with the lowest in-hospital mortality rates were those treated with GLP-1 RAs (0.4% due to COVID-19, and 0.2% for other reasons). Similar findings were observed for cumulative death over the two-year observation period. This trend persisted across all subjects, including those who did not require hospitalization within the two-year span, as well as those who experienced at least one hospitalization (Figure 1B,C).

Assessment of the length of stay, defined as the number of days spent during the primary hospitalization, revealed minimal disparities between the various subject groups. This observation held true even when comparing lengths of stay between COVID-19 and non-COVID-19 inpatients (refer to Appendix A).

### 3.4. One-versus-One Comparisons

One-versus-one analyses revealed notable differences when considering the risks associated with hospital admission, COVID-19-related hospitalization, in-hospital death, and cumulative death (refer to Figure 2). In terms of hospital admission, insulin/insulin secretagogues, GLP-1 RAs, and DPP-4 inhibitors (DPP-4is) exhibited higher risks when each category was compared to metformin (with ORs of 2.05 [95% CI 1.86–2.27], 1.67 [95% CI 1.25–2.22], and 2.23 [1.93–2.59], respectively). Moreover, DPP-4is displayed a significantly higher risk than GLP-1 RAs (with an OR of 1.34) in terms of hospital admission.

Regarding COVID-19-related hospitalization, the most significant comparison was between insulin/insulin secretagogues and metformin (with an OR of 0.74 [95% CI 0.60–0.92]), indicating a lower risk associated with metformin use.

The most substantial findings emerged from the one-versus-one analyses related to mortality; concerning in-hospital death, both insulin/insulin secretagogues and DPP-4is demonstrated lower levels of protection compared to metformin (with ORs of 2.82 [95% CI 2.32–3.42] and 3.19 [95% CI 2.45–4.15], respectively). Furthermore, DPP-4is exhibited a higher risk of mortality than GLP-1 RAs (with an OR of 2.60 [95% CI 1.30–5.19]). However, the comparison between GLP-1 RAs and metformin did not yield statistically significant results in terms of in-hospital death.

These analyses provide important insights into the relative risks associated with different antidiabetic treatments in the context of hospitalization and mortality outcomes.

Indeed, the analyses pertaining to cumulative death provide some of the most compelling insights. These analyses reaffirm what was presented in Figure 1, highlighting that insulin/insulin secretagogues and DPP-4 inhibitors (DPP-4is) continue to exhibit worse outcomes compared to metformin (with ORs of 3.19 [95% CI 2.99–3.41] and 2.65 [95% CI 2.40–2.92], respectively). Furthermore, DPP-4is, when compared to GLP-1 receptor agonists (GLP-1 RAs), continue to display an elevated risk of cumulative death (with an OR of 6.33 [95% CI 4.39–9.13]).

However, a distinct finding emerges from the one-versus-one analysis between GLP-1 RAs and metformin. This analysis yields an OR of 0.42 (95% CI 0.29–0.60, *p* < 0.001). This substantial OR signifies the powerful protective effect exerted by GLP-1 RAs against mortality over a two-year period.

### 3.5. Logistic Regression Analyses

Logistic regression analyses were conducted to assess the individual contribution of each variable in influencing the three selected outcomes: hospital admission, in-hospital death, and cumulative death. The reference categories chosen for comparison were as follows: female sex, age below 60 years for age categories, and metformin for antidiabetic treatments. Notably, female sex was found to be independently associated with lower ORs across all considered outcomes, indicating a lower risk. Conversely, higher ORs were observed for the variable “4-point MACE” in relation to all outcomes (as detailed in Table 2).

Insulin/insulin secretagogues and DPP-4 inhibitors (DPP-4is) displayed notably elevated and statistically significant ORs for all three outcomes examined. In contrast, GLP-1 receptor agonists (GLP-1 RAs) demonstrated comparable results only in relation to hospital admission and in-hospital death, with ORs of 1.69 (95% CI 1.41–2.52) and 1.84 (95% CI 1.13–4.42), respectively. However, no significant differences were observed concerning cumulative death when GLP-1 RAs were compared to metformin as the reference treatment.

### 3.6. Survival Estimates

The Mantel–Cox log-rank tests conducted over the course of the two-year observation period (Figure 3) revealed significant differences (*p* < 0.001) in terms of cumulative survival between subjects who were hospitalized and those who were not. As anticipated, non-hospitalized subjects exhibited markedly higher survival rates (Figure 3A). This initial observation prompted us to delve deeper into our investigation, wherein we sought differences among both the hospitalized and non-hospitalized subjects, while stratifying for different antidiabetic drugs.

Remarkably, GLP-1 receptor agonists (GLP-1 RAs) exhibited significantly better survival estimates (*p* < 0.001) for both cohorts of subjects throughout the entire observation period (Figure 3B,C). The Kaplan–Meier survival curves among hospitalized subjects displayed a consistent decline over time across all treatment groups. Conversely, for non-hospitalized subjects, the curves maintained a relatively stable trajectory up to around 300 days, after which they experienced a more rapid decrease across all treatment groups. Notably, non-hospitalized subjects treated with GLP-1 RAs demonstrated a higher likelihood of survival compared to those treated with other medications (with *p* < 0.05).

These findings persisted even when variables associated with COVID-19 and non-COVID-19-related hospitalizations were introduced into the analysis. Once again, GLP-1 receptor agonists (GLP-1 RAs) demonstrated significantly improved survival curves when compared to all other antidiabetic drugs (with *p*-values < 0.05) (Figure 3D–F).

## 4. Discussion

In addition to their hypoglycemic effects, DPP-4 inhibitors (DPP-4is) and GLP-1 receptor agonists (GLP-1 RAs) exert a broad anti-inflammatory influence. They achieve this by facilitating the transformation of blood and tissue monocyto-macrophagic cells into the anti-inflammatory M2 phenotype. Simultaneously, they decrease the production of inflammatory cytokines [15,20,21,22].

A recent study conducted on a total of 338 COVID-19 inpatients revealed that the administration of sitagliptin, a DPP-4 inhibitor, upon admission yielded substantial benefits. Among the group of 169 patients treated with sitagliptin, there was a significant reduction in in-hospital mortality when compared to the control group of 169 patients receiving conventional insulin treatment (18% vs. 37%; Hazard Ratio [HR] = 0.44). Notably, the use of sitagliptin was also associated with a reduced risk of mechanical ventilation and admission to intensive care units (ICUs), with Hazard Ratios (HRs) of 0.27 and 0.51, respectively [23]. In addition to these data and always in the context of COVID-19 inpatients, a multination meta-analysis showed that DPP-4is administration was associated with significantly reduced overall mortality rates (OR 0.75) [24].

GLP-1 RAs were considered excellent candidates for treating COVID-19 also in patients without diagnosis of T2DM owing to their multiple beneficial effects on excessive inflammation-induced acute lung injury [8], once again showing that COVID-19 can be considered a model of acute stressor against which molecules with a marked anti-inflammatory role can act.

A plausible explanation for the favorable effect of GLP-1 receptor agonists (GLP-1 RAs) on the clinical course of COVID-19 stems from an experimental study conducted in an animal model using streptozotocin-induced diabetes rats. This study demonstrated that liraglutide, a type of GLP-1 RA, can stimulate the expression of pulmonary ACE2 and Angiotensin (1-7) [A(1-7)], thereby reversing the imbalance within the renin-angiotensin system (RAS) in rats with type 1 diabetes mellitus (T1DM). This imbalance is characterized by a preponderance of the vasoconstrictor component of the RAS. This results in elevated levels of angiotensin II (AII), which subsequently leads to right ventricle hypertrophy. The study’s findings revealed that liraglutide effectively counteracted right ventricle hypertrophy and promoted increased production of proteins A and B of the pulmonary surfactant (SP-A and SP-B) in diabetic rats [25].

The role of ACE2 expression in COVID-19 pathogenesis was already hypothesized in many previous studies and its modulation was thought to be one of the keys for modulating the inflammatory response [26,27]. Moreover, it was also theorised that uncontrolled hyperglycemia may cause aberrant glycosylation of ACE2 in lungs, nasal airways, tongue, and oropharynx, thus increasing SARS-CoV-2 viral binding sites and leading to a higher trend of SARS-CoV-2 infections and more severe forms of COVID-19 [28]. For this reason, an efficacy regulation of plasma sugar levels plays a fundamental role in COVID-19 management.

The activation of the ACE2/A(1-7)/MasR axis is also able to determine an important antithrombotic effect [29,30], mediated by the production of prostacyclin and nitric oxide (NO) [31]. Furthermore, the restoration of the renin-angiotensin system (RAS) balance achieved by enhancing the activity of the ACE2/Angiotensin (1-7)/MasR axis has the potential to mitigate the pro-inflammatory state and suppress the excessive activation of the coagulation process. This modulation of the RAS can also help mitigate the development of thrombotic complications commonly associated with COVID-19, which is often referred to as COVID-19 coagulopathy [32,33], which plays a fundamental role in the pathogenesis of ARDS and multi-organ failure during SARS-CoV-2 infection [34] and is often associated with an ominous prognosis of COVID-19 [35]. Therefore, it is conceivable that the previously described effects of GLP-1 RAs on the synthesis of pulmonary surfactants proteins [25], may be able to determine a further protective effect on the COVID-19 clinical outcomes. Additionally, while the elevated expression of ACE2 might be assumed as a potential facilitator for SARS-CoV-2 cell entry, it is plausible that GLP-1 receptor agonists (GLP-1 RAs), through their counteraction of pro-inflammatory cytokine effects and restoration of RAS balance (including the enhancement of the ACE2/Angiotensin(1-7)/MasR axis activity), could potentially exert a protective effect against lung damage and the onset of multi-organ failure. In this context, GLP-1 RAs might contribute to reducing the severity of COVID-19 by mitigating the detrimental impact on lung function and overall organ health [9,36].

In the current state of research, numerous literature reports have emphasized the necessity for conducting clinical and epidemiological studies with the objective of evaluating the effects of GLP-1 receptor agonists (GLP-1 RAs) on the clinical outcomes of COVID-19 in patients with type 2 diabetes (T2DM) [8,9]. A recent retrospective observational clinical study highlighted that the prior use of GLP-1 RAs and SGLT-2 inhibitors (SGLT-2is), when compared to the utilization of DPP-4 inhibitors (DPP-4is), was linked to a substantial 60-day mortality reduction. Additionally, it was associated with noteworthy reductions in overall mortality, Emergency Room (ER) admissions, and hospitalizations [37]. Conversely, another retrospective observational study conducted in Denmark indicated that the utilization of incretin-based therapies was not associated with improved COVID-19 outcomes. However, it is worth noting that statistical power was constrained due to a small sample size [38].

A neutral effect of both incretin-based therapies (and SGLT-2is) on COVID-19-related mortality was also showed by a national retrospective observational study performed in England on a total of 2,851,465 T2DM patients [39].

Researchers from Indonesia conducted a study that revealed a significant association between the pre-admission use of GLP-1 receptor agonists (GLP-1 RAs) and a reduction in mortality rates related to COVID-19 among patients with type 2 diabetes (T2DM). The OR calculated for this association was 0.53, indicating a substantial reduction in the odds of mortality for those who were using GLP-1 RAs prior to their admission due to COVID-19. Importantly, this association held irrespective of other factors such as age, sex, pre-existing diagnosis of hypertension or other cardiovascular diseases, and the administration of other antidiabetic medications like metformin or insulin [40].

Two other recent meta-analyses have delved into the effects of preadmission use of antidiabetic medications on the in-hospital mortality of patients with type 2 diabetes (T2DM) and COVID-19. In the first meta-analysis, which encompassed 61 studies, it was found that preadmission use of certain antidiabetic medications correlated with distinct outcomes in terms of in-hospital mortality. Specifically, the use of metformin (OR 0.54), GLP-1 RAs (OR 0.51), and SGLT-2is (OR 0.60) was associated with lower mortality rates in individuals with diabetes and COVID-19. Conversely, the use of DPP-4is (OR 1.23) and insulin (OR 1.70) was linked to elevated mortality rates. Other antidiabetic medications such as sulfonylureas, thiazolidinediones, and alpha-glucosidase inhibitors exhibited a neutral impact on mortality outcomes [41]. The second meta-analysis showed instead that treatment with metformin (OR 0.74), DPP-4is (OR 0.88), SGLT-2is (OR 0.82), and GLP-1 RAs (OR 0.91) was related to reduced COVID-19 mortality rates in T2DM subjects, while insulin to increased mortality [42]. Additionally, GLP-1 RAs exhibited the most substantial and significant protective effect in reducing mortality rate, followed by SGLT-2is and metformin.

As for comparisons between the use of DPP-4is and GLP-1 RAs towards COVID-19 outcomes, conflicting results have been reported. While some studies have not shown a significant favourable effect on COVID-19 outcomes by DPP-4is [43,44], some others showed a possible protective action [45,46]. Moreover, while a recent meta-analysis performed on a total of 10 studies showed that DPP-4is therapy is not able to determine a significant improvement in COVID-19 outcomes [47], another meta-analysis which included the aforementioned study by Solerte et al. [23], in addition to 2 studies considering the intrahospital use of DPP-4is [46,48], highlighted a significant reduction in terms of mortality among T2DM patients treated with such drugs; the association was weaker in patients who were also taking metformin and/or ACE inhibitors [49]. A second recent meta-analysis by Indian scientists showed that intra-hospital administration of DPP-4is (pre-admission administration was not considered) was associated with significantly lower COVID-19-related mortality [50].

Based on the existing observational studies, it remains challenging to arrive at definitive conclusions regarding the impact of pre-existing DPP-4 inhibitor (DPP-4is) therapy on COVID-19 outcomes [51]. However, there is a plausible hypothesis that the continued use of incretin-based therapies within the hospital setting might have the potential to notably enhance clinical outcomes for COVID-19 patients. Therefore, it is crucial to conduct thorough analyses to investigate the potential effects of pre-admission treatment involving both DPP-4is and GLP-1 RAs on the clinical outcomes of COVID-19.

Even though the study did not specifically discuss treatment with SGLT-2 inhibitors (SGLT-2is), it is worth noting some recent research that highlights the anti-inflammatory effects of SGLT-2is. These medications have been shown to reduce the activity of pro-inflammatory cytokines such as tumor necrosis factor alpha (TNF-α), interleukin-6 (IL-6), and C-reactive protein (CRP) [52,53,54]. These effects are potentially mediated by SGLT-2is-induced reductions in uric acid, insulin levels [55], and leptin, as well as increases in adiponectin levels [56,57]. Additionally, SGLT-2is have demonstrated the ability to counteract low-grade inflammation and oxidative stress linked to diabetes [58], and they are associated with the polarization of monocyte-macrophage cells into the anti-inflammatory M2 phenotype [59,60].

Regarding sulfonylureas, a retrospective study conducted using the United Health Group Clinical Discovery Database revealed that both sulfonylureas and insulin were associated with increased odds of hospitalization among individuals with T2DM [61]. In contrast, a recent meta-analysis indicated that metformin and sulfonylureas might be linked to a reduced risk of mortality in patients with T2DM and COVID-19 [62]. Moreover, another recent meta-analysis corroborated the protective role of metformin against COVID-19-related deaths [63].

When assessing the effects of various antidiabetic medications on the clinical course of SARS-CoV-2 infection, it is important to note that there is a lack of definitive and conclusive data across the board. However, an exception to this trend appears to be metformin. Existing evidence suggests that metformin generally exerts a favorable effect, both in terms of reducing the risk of hospitalization and lowering mortality rates among individuals with SARS-CoV-2 infection [64,65,66,67]. A possible explanation of this phenomenon could involve the population of subjects with T2DM treated with metformin, the first line of treatment for T2DM and usually destinated to subjects with non-complicated forms of T2DM.

Our study demonstrates a favorable effect of GLP-1 RAs home therapy in the cohort of hospitalised and non-hospitalised subjects and also after a subgroups analysis concerning the subjects admitted for COVID-19 or for other reasons. Once again, we could state that GLP-1 RAs are able to offer an additional layer of protection for T2DM patients even under acute stressors like COVID-19, and this is in line with some of the aforementioned studies and meta-analyses [37,40,41,42]: in our cohort of patients the effects of GLP-1 RAs are evident in each step of the analyses performed, and it is linked to significantly lower percentage of death, even after a period of observation of two years.

The results of the logistic regression analyses performed clearly demonstrate how the burden of major adverse cardiovascular events (MACEs) in subjects with T2DM, coupled with their older age and male sex, directly correlates with a higher likelihood of hospital admission (for various reasons). This observation aligns well with the current literature and does not require further elaboration. Similarly, these same variables contribute to higher mortality rates in T2DM subjects, both during their hospital stay and cumulatively over time, leaving no room for misinterpretation. Simultaneously, the Mantel–Cox log-rank tests unambiguously reveal that both hospitalized and non-hospitalized T2DM subjects who achieve the best survival outcomes are those treated with GLP-1 RAs. This holds true for patients hospitalized due to a clear acute stressor, such as COVID-19, as well as those hospitalized for other medical reasons.

While many of the anti-inflammatory effects of GLP-1 RAs and DPP-4is still require further clarification, the current literature does not offer a consensus on the lasting benefits provided by these two categories of drugs. Additionally, a more extensive body of research dedicated to these antidiabetic agents could shed light on their potential to modulate the course of acute and/or chronic stressors to which patients with diabetes are particularly susceptible.

Our observations are subject to several limitations, primarily stemming from the retrospective nature of the study and variations in sample size among different subject cohorts. The use of current administrative data sources ensures immediate availability but comes with inherent limitations in terms of data variety. Accessing the necessary information would necessitate manual consultation of each patient’s electronic health record (EHR), which is both costly and time-consuming, hindering the analysis of comorbidities and the continuity of antidiabetic treatments during hospitalization. Furthermore, we consciously chose not to focus on certain other antidiabetic drugs (e.g., acarbose, SGLT-2is, thiazolidinediones), as their data were grouped under the umbrella category of “other drug combinations or other drugs”, a limitation that should also be acknowledged.

Additionally, important information regarding factors that could influence disease severity (such as medical history, level of physical activity, or laboratory results) is lacking due to the aforementioned reasons.

We recognize that drawing definitive conclusions from administrative data requires larger sample sizes and acknowledge the challenges in doing so. Nonetheless, we are confident that this study, akin to those exploring the effects of antidiabetic medications during acute stressors like COVID-19 or other medical conditions, can serve as a foundational step toward arriving at conclusive findings and a deeper understanding of these drugs.

## 5. Conclusions

In our cohort of individuals with T2DM, those receiving home treatment with GLP-1 RAs exhibit lower mortality rates compared to any other subgroup treated with various antidiabetic medications. Notably, favorable survival trends were consistently observed for both hospitalized individuals (for acute stressors such as COVID-19 or other medical conditions) and those who were not hospitalized. Furthermore, the beneficial impact of GLP-1 RAs appears to be enduring, resulting in enhanced cumulative survival among individuals with diabetes, even over a two-year observation period.

## Figures and Tables

**Figure 1 biomedicines-11-02292-f001:**
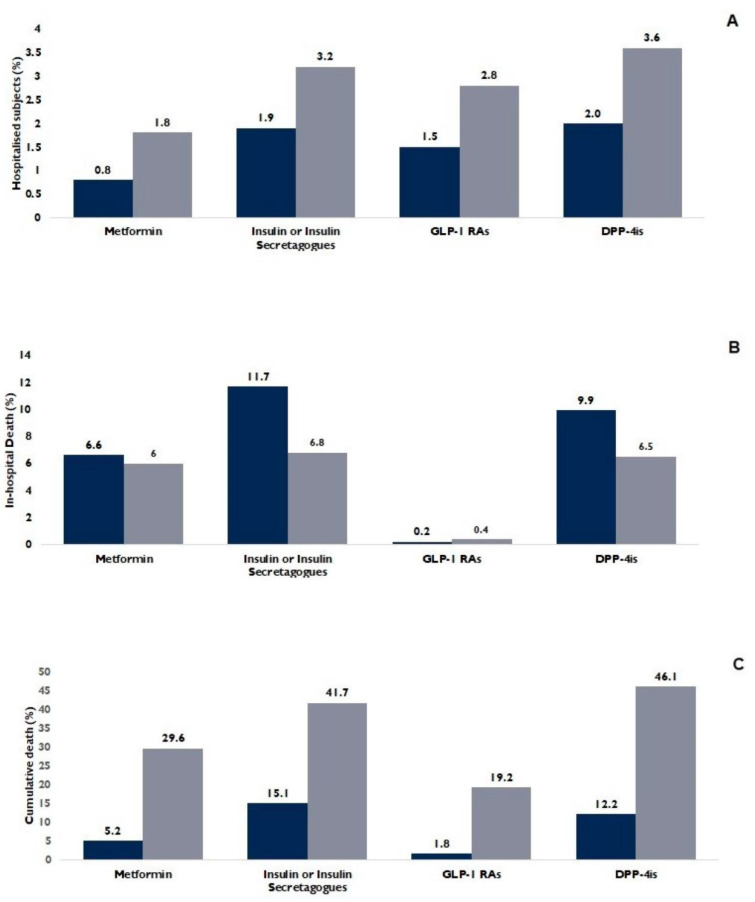
Comparison between the main four subgroups of diabetic inpatients in terms of hospital admission (admitted for COVID-19, right column, and for other reasons, left column) (**A**); Metformin vs. all drugs *p* < 0.001. Differences between groups in terms of in-hospital death (admitted for COVID-19, right column, and for other reasons, left column) (**B**); Metformin versus insulin/insulin secretagogues and versus DPP-4is. *p* < 0.001, GLP-1RAs vs. other drugs *p* < 0.001. Differences between groups in terms of cumulative death within the period of observation (admitted to hospital, right column, and not admitted to hospital, left column) (**C**). GLP-1 RAs vs. metformin and insulin/insulin secretagogues *p* < 0.001. DPP-4is vs. metformin and insulin/insulin secretagogues *p* < 0.005.

**Figure 2 biomedicines-11-02292-f002:**
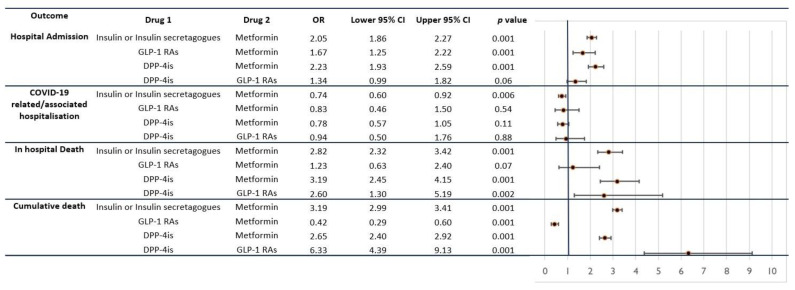
One-versus-one analyses evaluating the risk for worse outcomes (hospital admission, COVID-19-related/associated hospitalization, in-hospital death, and cumulative death). OR = odds ratio; 95% CI = 95% confidence interval; GLP-1 RAs = GLP-1 receptor agonists; DPP-4is = DPP-4 inhibitors.

**Figure 3 biomedicines-11-02292-f003:**
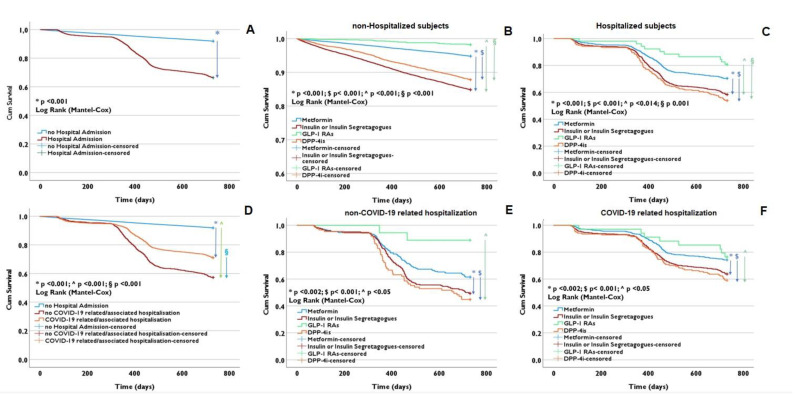
Mantel–Cox log-rank tests. Analysis of two-year survival among diabetic subjects subcohorts. Differences in terms of cumulative survival between hospitalised and non-hospitalised subjects (**A**); cumulative survival among non-hospitalised subjects based on antidiabetic treatment (**B**); cumulative survival among hospitalised subjects based on antidiabetic treatment (**C**); differences in terms of cumulative survival among non-hospitalised, hospitalised for COVID-19 and hospitalised for other reasons (**D**); cumulative survival among subjects hospitalised for other reasons based on antidiabetic treatment (**E**); cumulative survival among subjects hospitalised for COVID-19 based on antidiabetic treatment (**F**).

**Table 1 biomedicines-11-02292-t001:** Population characteristics.

Variables		Total	Non-Hospitalized Subjects	**Hospitalized Subjects (*n* = 2910)**		***p* Value**
				Non-COVID-19 Hospitalization	COVID-19 Hospitalization	
Subjects, n (%)		76,764	73,854 (96.2)	988 (34.0)	1922 (66.0)	<0.001
Age (years), mean ± SD		70 ± 13	70 ± 13	74 ± 13	73 ± 13	0.05
Age < 60 years, n (%)		14,861 (19.4)	14,435 (19.5)	135 (13.7)	291 (15.1)	0.36
Age 60–69 years, n (%)		18,291 (23.8)	17,789 (24.1)	164 (16.6)	338 (17.6)	0.57
Age 70–79 years, n (%)		24,278 (31.6)	23,372 (31.6)	288 (29.1)	618 (32.2)	0.23
Age ≥ 80 years, n (%)		19,334 (25.2)	18,258 (24.7)	401 (40.6)	675 (35.1)	0.05
Sex, n (%)	Female	35,418 (46.1)	34,231 (46.5)	410 (41.5)	777 (40.4)	<0.001
	Male	41,251 (53.9)	39,528 (53.5)	578 (58.5)	1145 (59.6)	
Days of hospital stay, mean ± SD		-	-	18.5 ± 16.3	18.2 ± 17.7	0.73
In-hospital death, n (%)		739 (1.0)	-	318 (32.2)	421 (21.9)	<0.001
Cumulative death, n (%)		6926 (9.0)	5947 (8.1)	423 (42.8)	556 (28.9)	<0.001
**C** **ardiovascular events**						
Non-fatal MI, n (%)		573 (0.7)	524 (0.7)	22 (2.2)	27 (1.4)	0.11
Non-fatal CVA, n (%)		1470 (1.9)	1277 (1.7)	81 (8.2)	112 (5.8)	0.023
Heart failure (HF), n (%)		1898 (2.5)	1589 (2.2)	125 (12.7)	184 (9.6)	0.022
Malignant dysrhythmia (MD), n (%)		323 (0.4)	287 (0.4)	14 (1.4)	22 (1.1)	0.53
Cardiac shock (CS), n (%)		50 (0.1)	40 (0.1)	0 (0.0)	10 (0.5)	0.019
4-point MACE, n (%)		3794 (4.9)	3263 (4.4)	215 (21.8)	316 (16.4)	0.004
**Antidiabetic drugs**						
Metformin, n (%)		30,238 (39.4)	29,455 (39.9)	239 (24.2)	544 (28.3)	0.07
Insulin or insulin secretagogues, n (%)		14,739 (19.2)	13,976 (18.9)	284 (28.7)	479 (24.9)	0.09
GLP-1 RAs, n (%)		1227 (1.6)	1175 (1.6)	18 (1.8)	34 (1.8)	0.92
GLP-1 RAs + metformin, n (%)		910 (1.2)	875 (1.2)	9 (0.9)	26 (1.4)	0.31
GLP-1 RA alone, n (%)		127 (0.2)	122 (0.2)	2 (0.2)	3 (0.2)	0.78
GLP-1 RAs + insulin or insulin secretagogues, n (%)		190 (0.2)	178 (0.2)	7 (0.7)	5 (0.3)	0.08
DPP-4is, n (%)		4301 (5.6)	4060 (5.5)	87 (8.8)	154 (8.0)	0.50
DPP-4i alone, n (%)		1169 (1.5)	1078 (1.5)	38 (3.8)	53 (2.8)	0.12
DPP-4is + metformin, n (%)		2037 (2.7)	1972 (2.7)	16 (1.6)	49 (2.5)	0.12
DPP-4is + insulin or insulin secretagogues, n (%)		1095 (1.4)	1010 (1.4)	33 (3.3)	52 (2.7)	0.35
Other drug combinations or other drugs, n (%)		26,259 (34.2)	25,188 (34.1)	360 (36.4)	711 (37.0)	0.84

Non-COVID-19 hospitalization, non-COVID-19-related/associated hospitalization but for other reasons; COVID-19 hospitalization, COVID-19-related/associated hospitalization; MI = myocardial infarction; CVA = cardiovascular accidents; HF = heart failure; CS = cardiac shock; MD = malignant dysrhythmia; 4-point MACE = 4-point major adverse cardiovascular events (non-fatal MI, non-fatal CVA, HF with or without CS, and MD); GLP-1 RAs = GLP-1 receptor agonists; DPP-4is = DPP-4 inhibitors. Data are presented as number (%), and if not are appropriately specified.

**Table 2 biomedicines-11-02292-t002:** Logistic regression analyses.

	B	S.E.	Wald	df	*p* Value	OR	95% CI (Upper-Lower)
**Hospital admission**							
Sex (F/M)	−0.27	0.05	29.63	1	<0.001	0.77	0.70–0.84
Age < 60 years	-	-	82.01	3	<0.001	-	-
Age 60–69 years	−0.01	0.09	0.01	1	0.99	1.00	0.84–1.18
Age 70–79 years	0.23	0.08	8.87	1	0.003	1.26	1.08–1.46
Age ≥ 80 years	0.55	0.08	51.96	1	<0.001	1.73	1.49–2.01
4-point MACE	1.31	0.07	396.51	1	<0.001	3.72	3.27–4.24
Metformin	-	-	115.75	3	<0.001	-	-
Insulin or insulin secretagogues	0.51	0.05	88.79	1	<0.001	1.66	1.50–1.85
GLP-1 RAs	0.64	0.15	18.37	1	<0.001	1.69	1.41–2.52
DPP-4is	0.58	0.08	55.97	1	<0.001	1.78	1.53–2.08
**In-hospital death**							
Sex (F/M)	−0.50	0.09	29.09	1	<0.001	0.61	0.51–0.72
Age < 60 years			179.07	3	<0.001	-	-
Age 60–69 years	0.67	0.30	4.88	1	0.03	1.95	1.08–3.53
Age 70–79 years	1.74	0.27	43.04	1	<0.001	5.68	3.38–9.53
Age ≥ 80 years	2.45	0.26	88.65	1	<0.001	11.62	6.97–19.35
4-point MACE	1.55	0.10	229.20	1	<0.001	4.71	3.85–5.75
Metformin			42.07	3	<0.001	-	-
Insulin or insulin secretagogues	0.59	0.10	33.33	1	<0.001	1.81	1.48–2.20
GLP-1 RAs	0.80	0.35	5.33	1	0.021	1.84	1.13–4.42
DPP-4is	0.67	0.14	23.59	1	<0.001	1.96	1.49–2.56
**Cumulative death**							
Sex (F/M)	−0.24	0.03	53.59	1	<0.001	0.79	0.74–0.84
Age < 60 years	-	-	2269.81	3	<0.001	-	-
Age 60–69 years	1.02	0.10	107.46	1	<0.001	2.76	2.28–3.35
Age 70–79 years	1.72	0.09	368.67	1	<0.001	5.57	4.67–6.64
Age ≥ 80 years	2.84	0.09	1066.26	1	<0.001	17.14	14.45–20.33
4-point MACE	1.16	0.05	578.85	1	<0.001	3.20	2.91–3.51
Metformin	-	-	565.10	3	<0.001	-	-
Insulin or insulin secretagogues	0.82	0.04	545.72	1	<0.001	2.28	2.13–2.44
GLP-1 RAs	−0.20	0.19	1.12	1	0.29	0.82	0.57–1.18
DPP-4is	0.60	0.05	124.76	1	<0.001	1.81	1.63–2.01

Logistic regression modelling for identifying the variables associated with outcomes (hospital admission, in-hospital death, and cumulative death). OR = odds ratio; 95% CI = 95% confidence interval; 4-point MACE = 4-point major adverse cardiovascular events (non-fatal MI, non-fatal CVA, HF with or without CS, and MD); GLP-1 RAs = GLP-1 receptor agonists; DPP-4is = DPP-4 inhibitors.

## Data Availability

Data are protected by a research consortium. The corresponding author will share the data upon receipt of a formal proposal from interested researchers.

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
