# Peer review of "The Impact of GLP-1 RAs and DPP-4is on Hospitalisation and Mortality in the COVID-19 Era: A Two-Year Observational Study"

_biomedicines, 2023, doi:10.3390/biomedicines11082292_

Round 1

Reviewer 1 Report

I received an original research article entitled "Impact of GLP-1 RAs and DPP-4is on Hospitalisation and Mortality in the COVID-19 Era: A Two-Year Observational Study" prepared by a team of researchers from Italy (Salvatore Greco et al.), which has been submitted to the journal Biomedicines (IF=4.7). The topic of the article, taken up by the Authors, is extremely important. Diabetes and its complications are one of the major public health problems in the world. Diabetes is an important risk factor favoring the occurrence of infection, and on the other hand, it is associated with an increased risk of a severe course of the disease. New antidiabetic drugs affecting the incretin system are considered a major breakthrough in diabetes therapy. It is worth emphasizing their pleiotropic effect, related, among others, to the immunomodulatory effect mentioned by the Authors. The COVID-19 pandemic has been one of the greatest challenges for the world's healthcare systems in this century. Although the main threat associated with this disease has already been averted, there are still cases with a severe course and unfavourable prognosis. All scientific reports that shed new light on this disease are valuable. On the other hand, the analysis of the impact of these drugs on the course of COVID-19 infection may allow for a better understanding of the immunomodulatory properties of these substances. In my opinion, the presented manuscript has a high substantive and scientific level and therefore should be considered for publication, but some corrections are needed, which I present below.

1)     In the description of the statistical analysis, the authors wrote "In the presence of symmetry of the distributions, the variables were represented with mean and Standard Deviation (SD)". Such a sentence is incorrect. The content of the sentence shows that the Authors considered that the symmetry of the distribution guarantees its normality. That's not true. Every normal distribution is symmetric, but not every symmetric distribution is normal. The normal distribution is a symmetric distribution with a strictly defined kurtosis. It should therefore be clearly stated that the mean and standard deviation were used in the case of compliance with the normal distribution.

2)     In all places in the text of the paper where the OR or HR value is given, the 95% CI value should also be given.

3)     Limitations of the study should be discussed in more detail. In my opinion, one of the most serious limitations is the lack of consideration of other factors that may affect the severity of the disease (see, for example, recent studies on the relationship between physical activity and the severity of the disease – 10.3390/jcm12124046).

4)     Minor grammatical and stylistic errors should be corrected.

Minor editing of English language required

Author Response

I received an original research article entitled "Impact of GLP-1 RAs and DPP-4is on Hospitalisation and Mortality in the COVID-19 Era: A Two-Year Observational Study" prepared by a team of researchers from Italy (Salvatore Greco et al.), which has been submitted to the journal Biomedicines (IF=4.7). The topic of the article, taken up by the Authors, is extremely important. Diabetes and its complications are one of the major public health problems in the world. Diabetes is an important risk factor favoring the occurrence of infection, and on the other hand, it is associated with an increased risk of a severe course of the disease. New antidiabetic drugs affecting the incretin system are considered a major breakthrough in diabetes therapy. It is worth emphasizing their pleiotropic effect, related, among others, to the immunomodulatory effect mentioned by the Authors. The COVID-19 pandemic has been one of the greatest challenges for the world's healthcare systems in this century. Although the main threat associated with this disease has already been averted, there are still cases with a severe course and unfavourable prognosis. All scientific reports that shed new light on this disease are valuable. On the other hand, the analysis of the impact of these drugs on the course of COVID-19 infection may allow for a better understanding of the immunomodulatory properties of these substances. In my opinion, the presented manuscript has a high substantive and scientific level and therefore should be considered for publication, but some corrections are needed, which I present below.

1)     In the description of the statistical analysis, the authors wrote "In the presence of symmetry of the distributions, the variables were represented with mean and Standard Deviation (SD)". Such a sentence is incorrect. The content of the sentence shows that the Authors considered that the symmetry of the distribution guarantees its normality. That's not true. Every normal distribution is symmetric, but not every symmetric distribution is normal. The normal distribution is a symmetric distribution with a strictly defined kurtosis. It should therefore be clearly stated that the mean and standard deviation were used in the case of compliance with the normal distribution.

- That is true. We changed the sentence with the following: “In case of normal distribution of data, continuous variables were presented with mean and Standard Deviation (SD), while in case of non-normal distribution, with median value and interquartile range [1Q 3Q].”

2)     In all places in the text of the paper where the OR or HR value is given, the 95% CI value should also be given.

- The 95% Confidence Intervals were added in all paragraphs concerning our study.

3)     Limitations of the study should be discussed in more detail. In my opinion, one of the most serious limitations is the lack of consideration of other factors that may affect the severity of the disease (see, for example, recent studies on the relationship between physical activity and the severity of the disease – 10.3390/jcm12124046).

- This sounds right. We added this statement in the limitations section “Additionally, important information regarding factors that could influence disease se-verity (such as medical history, level of physical activity, or laboratory results) is lack-ing due to the aforementioned reasons.”

4)     Minor grammatical and stylistic errors should be corrected.

- We promptly revised them.

On behalf of all authors, We want to thank the reviewer for his/her kind and valuable comments. We really appreciate the work behind each of them.

Reviewer 2 Report

The manuscript "Impact of GLP-1 RAs and DPP-4is on Hospitalisation and Mortality in the 2 COVID-19 Era: A Two-Year Observational Study, by Greco S et all, is a very important study in the field of diabetes, especially for patients with Type 2 diabetes and COVID. Several studies and many reviews and meta-analyses were made regarding the potential benefits of antidiabetic therapy for patients with diabetes and COVID. The anti-inflammatory effect of GLP1RA is well known but it is important to understand if this effect could be translated also for patients with COVID.

This manuscript enrolled a large number of patients and respected the ethical requirements. The statistical analyses were very well made and the discussions and conclusions are in line with the main findings. The scientific soundness is very well and overall, the manuscript is qualified for publication.

Author Response

Thanks for your kind response and your review.

We really appreciate your comments and the work behind them.

Reviewer 3 Report

The authors present an observational cohort study aiming to explore the potential protective role of GLP-1 RAs and DPP-4is in both COVID-19 patients (as a model of acute stressor) and non-COVID-19 patients, all of which are under a two-year observation period.

 Comments:

1. 

The Abstract is suggested to include effect size measurements in the results analysis, such as odds ratios (OR) (95% confidence interval [CI]) or hazard ratios (HR) (95% CI).

2.

Figure 1 displays the distribution of antidiabetic therapy, and Table 1 presents the same information. However, Table 1 is missing the 'GLP-1 RA+metformin, n=910' group. If the information presented is redundant, please choose one for reporting.

3.

Is Figure 3 a univariate logistic regression?

Is Table 2 a multivariate logistic regression?

Since the odds ratios (OR) (95% confidence interval) in Figure 3 and Table 2 are different.

4.

Apart from the analysis of COVID-19-related hospitalization, which was not included in Table 2, and the comparison between DPP-4 inhibitors and GLP-1 RAs, the study explored the outcomes related to hospital admission, in-hospital mortality, and cumulative death in relation to the exposure factor of antidiabetic drugs.

Can Figure 3 and Table 2 be integrated into a single Table or Figure?

For Table 2, it is suggested to present the odds ratios (OR) (95% confidence interval) for univariate or multivariate logistic regression.

The presentation of B, S.E., Wald, and df may be omitted.

5.

Additionally, the statistical methods employed were binary or multinomial Logistic Regression. Regarding the statement 'Significant differences were evidenced in one-vs-one analyses considering the risks for COVID-19 related hospitalization,' does its outcome include COVID-19 hospitalization, Non-COVID-19 hospitalization, and Non-hospitalized subjects?

Binary logistic regression is used when the dependent variable has two categories, while multinomial logistic regression extends the approach for situations where the independent variable has more than two categories.

4.

Is it a TYPO to use 'Mantel-Cox long-rank tests' in Figure 4, or should it be 'Mantel-Cox log-rank tests'?

5.

Figure 4 presents Mantel-Cox log-rank tests, which is a statistical method for a cohort study design, while Figure 3 and Table 2 display odds ratios (OR), akin to a case-control study design's statistical approach.

If a cohort study design is intended, it is suggested to use Cox proportional hazards regression model, especially for Figure 3 and Table 2, with the effect size presented as hazard ratios (HR) (95% confidence interval).

6.

Figure 4 presents Mantel-Cox log-rank tests, which is a statistical method for a cohort study design, while Figure 3 and Table 2 display odds ratios (OR), similar to a case-control study design's statistical approach.

If a cohort study design is intended, it is suggested to utilize Cox proportional hazards regression model, especially for Figure 3 and Table 2, with its effect size presented as hazard ratios (HR) (95% confidence interval).

As a result, the all-cause mortality data from Figure 2 can also be integrated within the consolidated Figure 3 and Table 2.

This approach would enable the presentation of not only the percentage of cumulative events but also the incidence rate/mortality rate per 1000 Person-Years.

7.

A concise presentation of figures and tables, along with appropriate statistical methods, is essential for this research article.

Author Response

The authors present an observational cohort study aiming to explore the potential protective role of GLP-1 RAs and DPP-4is in both COVID-19 patients (as a model of acute stressor) and non-COVID-19 patients, all of which are under a two-year observation period.

 Comments:

  1.  

The Abstract is suggested to include effect size measurements in the results analysis, such as odds ratios (OR) (95% confidence interval [CI]) or hazard ratios (HR) (95% CI).

  • We believe that including ORs, HRs and 95% CIs would uselessly complicate the read. Since many numbers and percentages have been included in the manuscript, the reader can find them in the appropriate section. In the abstract we reported only our conclusion in order to make it easier to browse.

2.

Figure 1 displays the distribution of antidiabetic therapy, and Table 1 presents the same information. However, Table 1 is missing the 'GLP-1 RA+metformin, n=910' group. If the information presented is redundant, please choose one for reporting.

  • Figure 1 was removed, and all the information presented in figure 1 and table 1 was aggregated in table 1. We also added the missing information concerning the group of patients treated with GLP1-RAs + metformin in table 1. Thanks for pointing this out.

3.

Is Figure 3 a univariate logistic regression?

Is Table 2 a multivariate logistic regression?

Since the odds ratios (OR) (95% confidence interval) in Figure 3 and Table 2 are different.

  • Figure 3 (now figure 2) contains one-vs-one analyses and thus univariate regressions, while table 2 multivariate regressions. That is why the ORs are different.

4.

Apart from the analysis of COVID-19-related hospitalization, which was not included in Table 2, and the comparison between DPP-4 inhibitors and GLP-1 RAs, the study explored the outcomes related to hospital admission, in-hospital mortality, and cumulative death in relation to the exposure factor of antidiabetic drugs.

Can Figure 3 and Table 2 be integrated into a single Table or Figure?

For Table 2, it is suggested to present the odds ratios (OR) (95% confidence interval) for univariate or multivariate logistic regression.

The presentation of B, S.E., Wald, and df may be omitted.

  • In table 2 we voluntarily omitted the outcomes related to COVID-19. Since Figure 3 (now Figure 2 after deleting Figure 1) and table 2 refer to different aims and analyses, we believe that they cannot be integrated in a single table/figure. As for B, S.E., Wald and df, they could represent a possibility to better understand the findings of these regression analyses, especially for those readers who are not experts in the field.

5.

Additionally, the statistical methods employed were binary or multinomial Logistic Regression. Regarding the statement 'Significant differences were evidenced in one-vs-one analyses considering the risks for COVID-19 related hospitalization,' does its outcome include COVID-19 hospitalization, Non-COVID-19 hospitalization, and Non-hospitalized subjects? Binary logistic regression is used when the dependent variable has two categories, while multinomial logistic regression extends the approach for situations where the independent variable has more than two categories.

  • In one-vs-one analyses we compared each variable with a single other variable within the same category, generating odds ratios for each individual outcome in turn. For example, regarding hospital admission, GLP1-Ras had an odds ratio of 1.67 compared to metformin with a p-value < 0.001.

6.

Is it a TYPO to use 'Mantel-Cox long-rank tests' in Figure 4, or should it be 'Mantel-Cox log-rank tests'?

  • It was a mistake. Thanks for noticing it, we corrected it.

7.

Figure 4 presents Mantel-Cox log-rank tests, which is a statistical method for a cohort study design, while Figure 3 and Table 2 display odds ratios (OR), akin to a case-control study design's statistical approach.

If a cohort study design is intended, it is suggested to use Cox proportional hazards regression model, especially for Figure 3 and Table 2, with the effect size presented as hazard ratios (HR) (95% confidence interval).

  • That's a valid observation. However, we have decided to choose the log-rank test over Cox regressions in order to simplify the interpretation of the survival curves, especially considering the absence of additional covariates to introduce in the analysis. This way, given the need to create straightforward survival curves without any confounding variables, the log-rank test does not show any inferiority compared to Cox regressions. Once again, since Figure 3 (now Figure 2) and table 2 present extremely different parameters, they cannot be summarized in a single figure/table.

8.

Figure 4 presents Mantel-Cox log-rank tests, which is a statistical method for a cohort study design, while Figure 3 and Table 2 display odds ratios (OR), similar to a case-control study design's statistical approach.

If a cohort study design is intended, it is suggested to utilize Cox proportional hazards regression model, especially for Figure 3 and Table 2, with its effect size presented as hazard ratios (HR) (95% confidence interval).

As a result, the all-cause mortality data from Figure 2 can also be integrated within the consolidated Figure 3 and Table 2.

This approach would enable the presentation of not only the percentage of cumulative events but also the incidence rate/mortality rate per 1000 Person-Years.

  • The answer to this concern was given above, since the point n.7 and 8. Are practically comparable

  1.  

A concise presentation of figures and tables, along with appropriate statistical methods, is essential for this research article.

  • We tried to follow all your suggestions for this work. As for some of your concerns, we could not fully do it, even if we tried to clarify some things that could probably sound ambiguous.

On behalf of all authors, we would like to sincerely thank the reviewer for his/her kind and professional contribution to our article. We really appreciate the work behind this.

Round 2

Reviewer 1 Report

The paper has been significantly improved. I have no further comments. I recommend it for publication in its current form.

Reviewer 3 Report

None